# Development of a Logic Model for a Programme to Reduce the Magnetic Resonance Imaging Rate for Non-Specific Lower Back Pain in a Tertiary Care Centre

**DOI:** 10.3390/healthcare9020238

**Published:** 2021-02-23

**Authors:** Ahmed Alhowimel, Faris Alodaibi, Mazyad Alotaibi, Dalyah Alamam, Hana Alsobayel, Julie Fritz

**Affiliations:** 1Department of Health and Rehabilitation Science, Prince Sattam Bin Abdulaziz University, Al-Kharj 16278, Saudi Arabia; maz.alotaibi@psau.edu.sa; 2College of Applied Medical Sciences, Health Rehabilitation Sciences, King Saud University, Riyadh 11451, Saudi Arabia; falodaibi@ksu.edu.sa (F.A.); dalimam@ksu.edu.sa (D.A.); hsobayel@ksu.edu.sa (H.A.); 3Department of Physical Therapy and Athletic Training, University of Utah, Salt Lake City, UT 84112, USA; julie.fritz@utah.edu

**Keywords:** lower back pain, logic model, magnetic resonance imaging

## Abstract

Tertiary care centres continue to experience over-utilisation of diagnostic imaging services for lower back pain cases that may not be required. Moreover, these services may require additional time and consequently delay access to services that offer conservative management, i.e., physiotherapy, and hence, increase the direct and indirect costs with no added quality of care. A logic model was developed based on qualitative and quantitative studies that explains the plan and process evaluation strategies to reduce imaging for lower back pain in tertiary hospitals. Logic models are useful tools for defining programme components. The delivery of the components is ensured by well-defined process evaluations that identify any needed modifications. The proposed logic model provides a road map for spine clinics in tertiary care hospitals to decrease the number of patient referrals for magnetic resonance imaging and waiting times for consultations and services and promote early access to physiotherapy services.

## 1. Highlights

Early access for lower back pain (LBP) for physiotherapy treatment will result in better patient’s reported outcomes.Successful implementation of this logic model will result in efficient utilisation of spine clinic.

## 2. Introduction

Although there is consensus in international guidelines against the use of imaging for routine diagnostic tests of lower back pain [1,2,3,4,5,6], several studies have identified an increase in imaging referrals for diagnostic testing [7,8,9]. This issue clearly indicates a gap and poor adherence to the established guidelines amongst clinical practitioners. Apart from the direct cost of imaging, diagnosing people with lower back pain (LBP) using magnetic resonance imaging (MRI) has received a considerable amount of criticism. For example, some studies [10,11] have concluded that MRI lacks the capacity to identify the primary pathology, and others have reported that imaging unnecessarily exposes patients to radiation [12]. Furthermore, MRI is prone to yield false-positive findings, resulting in a larger number of patient referrals to specialty facilities. The results of a recent review of qualitative research [13] support the claim that using MRI to diagnose patients with chronic lower back pain (CLBP) is problematic. The patients in that study described their experiences of high levels of anxiety and fear regarding their diagnoses and prognoses after undergoing MRI. Moreover, patients might experience an elevated sense of catastrophising, resulting in avoidance and constrained movements to avoid pain exacerbation. Therefore, patients might be attached to the imaging report and require more imaging to monitor any symptoms.

A promising programme for reduction in imaging rates in spine clinics trained physiotherapists to triage new patients and refer only those with clear indications (e.g., chronic steroid use, the presence of neurological deficits) of the need for surgery to a spine surgeon. Patients who exhibited no indications of the need for spinal surgery received relevant education and self-management advice. In addition, patients were triaged using the STarT Back Screening Tool, as it provides the most appropriate guideline for recommending subsequent treatment. The preliminary data showed promising results and reductions in referrals for imaging and wait times for appointments with surgeons [12].

Patients’ expectations might also inform diagnostic and treatment options. The MRI’s lack of objective evidence supporting patients’ subjective symptoms and certain diagnosis could be unacceptable to them [14]. On the other hand, several studies have found that patients may experience feelings of relief when they receive a diagnosis, even if the diagnosis does not explain their pain. One study [15] examined the effect of diagnostic labelling on patients’ self-management and pain-related guilt experiences. Although the patients with CLBP experienced a greater sense of relief following an MRI, they also experienced heightened feelings of guilt regarding their apparent lack of improvement [14].

The provision of alternative treatment options for patients has been suggested as a way to overcome issues of acceptability, and physiotherapy services are known to be credible and acceptable options for many patients [16,17,18]. Therefore, to overcome patient expectations, it is recommended to provide physiotherapy treatment as a first line of treatment instead of providing unnecessary imaging. Moreover, a recent feasibility study about the acceptability of providing physiotherapy as an alternative to MRI in Saudi Arabia showed both staff and patient acceptability to the approach [19].

A considerable number of studies suggest that rapid intervention with physiotherapy for LBP is associated with better patient outcomes and greater patient satisfaction [20,21,22]. Moreover, the direct costs associated with LBP—including the number of MRI scans, visits to a general practitioner and prescription medications—were lower for people with CLBP who received early physiotherapy intervention [20,21,22,23].

This evidence indicates that early access to physiotherapy treatment may prevent unnecessary medical expenditures related to the use of imaging, medication and consultation whilst also improving the degree of patient self-efficacy and satisfaction. However, delayed access to physiotherapy for patients with LBP is associated with poor outcomes (i.e., physical disability, days lost of work and levels of satisfaction) and increases in chronicity [24,25,26].

## 3. A Logic Model for Decreasing the MRI Rate of Persons with CLBP in Saudi Arabia

### 3.1. Development

Logic modelling has been described as a practical way of representing theoretical data by presenting graphical/textual information on how a programme might work [27]. Ideally, the development of a logic model incorporates critical thinking by all the targeted stakeholders, or “resources”; it is highly dynamic by nature and can be enhanced at any stage of its implementation [27]. The logic model presented here is based mainly on the findings of a previous research conducted in a Saudi Arabian tertiary care setting, which involved qualitative interviews with patients, physiotherapists and spine surgeons [13], as well as quantitative analyses to test the feasibility and acceptability of using the expertise of a physiotherapist to screen all patients referred to a spine clinic [19]. The model described here includes the strategies for reducing MRI rates that were implemented successfully in the Saskatchewan Spine Pathway Study conducted in Canada [28]. 

### 3.2. The Context

The current care pathway in Saudi Arabia indicates that patients with LBP are referred to tertiary care spine clinics mainly from primary care centres. Patients in primary care receive no physiotherapy interventions because of the lack of such services at the primary care level and the fact that physiotherapy services in tertiary care are referral based. Delayed access to physiotherapy treatment is associated with more costs and increased chronicity [24,25]. Consequently, time away from work may increase and have a negative effect on the work productivity of patients with LBP [25]. Poor utilisation of the healthcare system might, therefore, have more severe socioeconomic consequences for individuals and society.

### 3.3. Determinants 

The proposed logic model programme will be implemented as a pilot trial in one spine clinic. Several factors might affect successful implementation of this project in private physiotherapy clinics. We have identified 16 factors related to characteristics of the individual, outer setting, inner setting, process and characteristics of the intervention (Table 1).

Furthermore, of these detriments, two potential challenges should be anticipated before the implementation of the programme. First, this is the first programme to address this issue in Saudi Arabia, and its implementation will require a qualified physiotherapist, which might impose a staff shortage in the physiotherapy department of the tertiary care centre. Second, there is a need to authorise physiotherapists working in the spine clinic to refer patients to various specialists, which might be hampered by administrative resistance because of the novelty of this privilege.

### 3.4. The Problem

According to the findings of the qualitative study conducted at the tertiary care centre [11], healthcare practitioners raised concerns about over-utilisation of MRIs for patients with LBP; they also reported that scanning was performed routinely for most of the patients with LBP in the tertiary care centre. The routine use of MRIs can cause patients to depend on their radiological diagnoses and request follow-up MRI scanning for monitoring and surveillance. Treatment with physiotherapy as early as possible, compared to not having early physiotherapy, is associated with better patient outcomes and greater patient satisfaction [20,21,22]. Moreover, a decrease in the cost of care has been documented [20,21,22,23].

### 3.5. The Evidence-Based Intervention

The results of the implementation of physiotherapy as a first option in the Saskatchewan Spine Pathway clinic were promising [28]. They showed a marked decrease in MRI referrals and reduced surgical candidates’ waiting times for consultation appointments with a spine surgeon. Another successful Canadian implementation programme showed similar results while maintaining a patent satisfaction rate of 97% [29,30]. Both of these implementation programmes yielded marked reductions in the cost of care for patients with LBP. Therefore, the proposed logic model would promote disseminating evidence based on back pain treatment pathways that would assist local service planning, performance monitoring and outcome evaluation.

### 3.6. Proposed Study Objectives

Decrease the rate of MRI referrals by providing early physiotherapy for LBP without serious pathology signs and symptoms.

Promote patients’ accessibility to spine and physiotherapy services.Screening patients who present no red flag and early referral to physiotherapy will reduce the waiting time for patients with need for spine surgical intervention.Achieve higher levels of satisfaction among patients and staff.Early studies in Saudi Arabia showed an acceptability to physiotherapy referals without having MRI imaging [19]. Therefore, we aim to achieve high satisfaction among patients and healthcare workers with implementation of this program.

### 3.7. Proposed Study Sitting

The target population in the programme is people with LBP referred from primary care centres to the spine surgery clinic of a tertiary care centre in Saudi Arabia. The proposed programme will initially be implemented in a spine surgery clinic of a tertiary care centre in Saudi Arabia. At this stage, we will not consider applying the project in other clinics. Following its successful implementation, the results will be presented to the Ministry of Health of Saudi Arabia to seek implementation of this programme in all tertiary care hospitals in Riyadh within one year, and the surrounding areas in Saudi Arabia within three years. The program can also be implemented in other clinics serving LBP patients such as orthopaedic, neurosurgery, rheumatology and neurology clinics.

### 3.8. Proposed Study Management Team

The primary investigator of this project is A.A., a physiotherapist with a PhD in rehabilitation, who has conducted several studies in Saudi Arabia about the current care for people with LBP. The leading staff from both the spine surgery clinic and physiotherapy clinic will be responsible for: first, gaining authorisation from the higher management to launch the study. Secondly, identifying the manpower needed to operate the program for six months. The educational component of this study will be led by F.A. and J.F., both physiotherapists with PhDs, with extensive experience in LBP management and differential diagnosis.

### 3.9. Partners and Stakeholders

It is essential for the physiotherapy department to work in partnership with the spine clinic for this programme to be successful. Joint stakeholder engagement is crucial for planning the implementation and arranging the education and the staffing of the department. Additionally, hospital administration involvement is a key in providing data before program initiation and approving the program.

### 3.10. Process Evaluation Objectives

All musculoskeletal (MSK) physiotherapists will complete training to screen for risk factors for serious pathology (i.e., red flags) that require referrals to other services and for psychosocial risk factors (i.e., yellow flags). Physiotherapists have the competencies necessary to identify patients who present with serious pathology [31,32,33,34] and psychosocial impairments [35,36,37] and to refer patients to appropriate medical services.Trained physiotherapists will receive training to use the software required to make referrals in the electronic referral system.Trained physiotherapists and spine surgeons will complete additional training in their inter-professional education and collaborative practice. Despite the belief held by healthcare providers that they work in collaboration with other providers, they might use their skills to achieve a common goal but might not be fully aware of the clinical skills and roles of the other team members involved in the patient care. This lack of awareness may lead to the under-utilisation of other healthcare providers’ skills and capabilities, which could have a negative effect on the quality of care. Inter-professional education is an interactive learning experience of providers from more than one healthcare profession, which offers opportunities to improve the collaborative process and, eventually, patient care [36].After completing the aforementioned training, a physiotherapist will evaluate all patients on their first visit to the spine clinic and triage them (if needed) as candidates for surgery and consultation with a spine surgeon. The remaining patients will be screened for psychosocial risk factors using the STarT Back Screening Tool and subsequently referred to the appropriate service, i.e., physiotherapy or a psychologically informed physiotherapy intervention (see Figure 1).Prior to referral to the physiotherapy outpatient department, all patients will be educated by the screening physiotherapist about ways to manage their back pain and the importance of staying active. Patients identified as being at high or medium risk, according to the STarT Back Screening Tool, will receive psychosocial support in the form of a combined physical and psychologically informed physiotherapy intervention provided by the physiotherapy department.

### 3.11. Proposed Study Data Collection and Outcome Measures

A number of demographic data will be collected from patients at the beginning of the programme (Table 1). Additionally, all patients will fill in a screening form for psychosocial risk factor at the bassline (STarT Back Screening Tool).

Moreover, four patient-reported outcomes will be collected at baseline and at discharge of every patients. These outcomes include, pain (using VAS) [38], disability (using Roland Morris disability scale) [39], self-efficacy using (pain self-efficacy scale) [40] and quality of life (using MSK-HQ) [41].

### 3.12. Process Outcomes

The target percentages were estimated based on implementation studies conducted in Canada [28,29,30].

By the end of the pilot (6 months), there will be a 25% reduction in MRI referrals.By the end of the pilot (6 months), the waiting time to see a spine surgeon will decrease by 30%.By the end of the pilot (6 months), the accessibility of physiotherapy interventions will increase 30%.A clinically significant decrease in disability and pain outcomes.A clinically significant increase in self-efficacy and quality of life outcomes.By the end of the pilot (6 months), 90% of the patients will report a high level of satisfaction with the clinic.

### 3.13. Strategies and Activities

The overall aim of this logic model is to reduce the rate of unnecessary MRI services and to promote rapid access to physiotherapy and spine surgery services whilst maintaining a high level of patient satisfaction (Table 2).

### 3.14. First Strategy

To provide training for MSK physiotherapists to screen patients to identify candidates for surgery, other medical services and rapid physiotherapy.

This step is the cornerstone of the programme, as early identification of surgery candidates can reduce their waiting times for consultation from a spine surgeon. Moreover, early referrals for physiotherapy have been found to increase positive patient outcomes. The suggested method for testing this strategy is to survey MSK physiotherapists before and after the training. The survey will include clinical scenarios of patients with LBP and questions about the appropriate decisions to make as the “first point of contact physiotherapist”. Focus groups and individual interviews with the trainees will also be conducted to explore the need for additional training and to identify obstacles the “first point of contact physiotherapist” screener might encounter in performing this role.

### 3.15. Second Strategy

We aim to provide training for MSK physiotherapists and spine surgeons on inter-professional education and collaborative practice. Without adequate and appropriate communication and a shared understanding of the programme’s purpose and potential outcomes, in terms of patients’ benefits (i.e., including better access to appropriate services) and decreased workloads for other healthcare workers, the programme might fail. Similar methods of testing could be implemented to measure the awareness of trainees about the need for multidisciplinary teamwork and effective interdisciplinary communication.

### 3.16. Third Strategy

To provide training for MSK physiotherapists and spine surgeons on best practice recommendations for educating patients with LBP, as recommended in evidence and recent guidelines, including guideline recommendations on the role of diagnostic imaging. All patients visiting the clinic should have an evidence-based educational programme based on the recommendations made by the screening physiotherapist, which mainly focus on providing assurance and promoting self-management.

## 4. Evaluation

### 4.1. Initial Evaluation

***Rationale*:** A formative evaluation is necessary to collect baseline data for comparisons with post-implementation data on the programme’s impact, which will be measured later in the evaluation process.

***Methods and indicators*:** Before the programme starts, it is important to know the number of physical therapists that will be working at the screening clinic based on the number of patients the clinic sees per day and the number of hours/day the clinic operates. This step will determine the staffing needs of the physiotherapy department. It is also important to retrieve from the spine clinic records from the previous 6 months on the following indicators for comparison purposes:Number of LBP patients referred to the spine surgeon and time to access the service;Number of LBP patients referred to other specialists;Percentage and number of LBP patients referred to physiotherapy by spine surgeon and time to access the service;Percentage and number of LBP patients referred for MRIs.

Moreover, as part of the programme’s evaluation, data will be collected for the previous 6 months regarding the cost of care associated with LBP, specifically health service utilisation, mainly for MRI scans.

### 4.2. Process Evaluation

***Rationale:*** A process evaluation is necessary to assess whether the activity outputs outlined in the logic model are contributing to the goals. Programme monitoring and data collection on activity outputs will be implemented throughout the duration of the project. Indicators will be used to monitor the achievement of outcomes.

***Methods and indicators*:** One method of examining how the training provided to the MSK physiotherapists reflected in their practice is to survey the therapists about relevant clinical practices using clinical scenarios adapted from the training sessions and by measuring the therapists’ awareness of red and yellow flags, e.g., their signs and symptoms. The following indicators will be used to ensure the practice of screening in the spine clinic is performed as planned:Percentage of patients referred to a spine surgeon.Percentage of patients referred to other specialists.Percentage of patients referred for physiotherapy.Percentage of patients referred for MRIs.Satisfaction level of the patients.

### 4.3. Impact Evaluation

The overall impact of this implementation will be reflected in the decreased rate of MRI scans for people with LBP and reduced number of days to access the service. Furthermore, it will increase patient-reported outcome of pain, disability and quality of life. Additionally, cost of care will be reduced. A comparison of MRI costs, medications and medical visits will be analysed as part of a separate evaluation before and after the programme.

## Figures and Tables

**Figure 1 healthcare-09-00238-f001:**
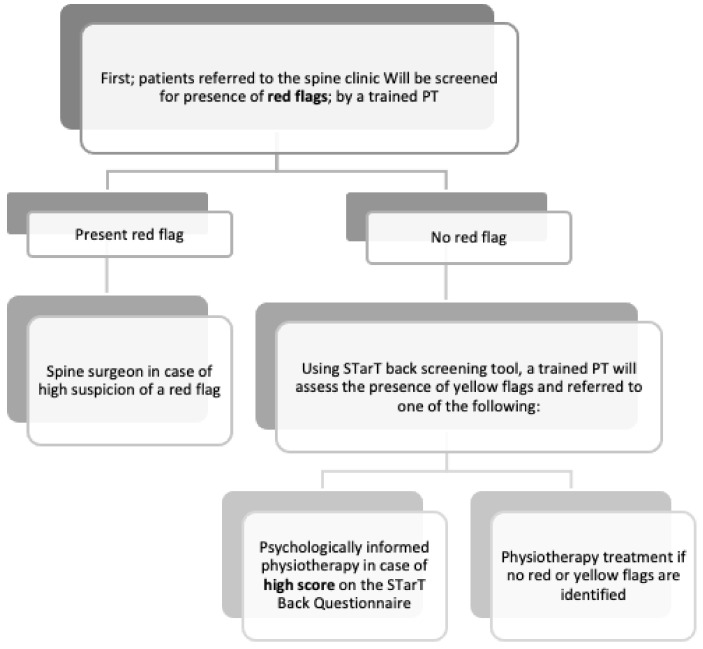
Screening process.

**Table 1 healthcare-09-00238-t001:** Bassline demographics and patient-reported outcome measures.

Demographics and Disease-Related Characteristics	Baseline	At Discharge
Sex (M/F), no. (%)	√	
Age (years)	√	
Lower back pain duration (months)	√	
Smoking (current (average per week)/previous (average per week)/never)	√	
Diabetes (y/n)	√	
Cardiovascular disease (y/n)	√	
Dyslipidaemia (or treatment for this) (y/n)	√	
Mental disorder (depression, anxiety) (y/n)	√	
Height (cm)	√	
Weight (kg)	√	
Patient-reported outcome measures		
Visual Analog Scale, VAS (0–10)	√	√
Roland Morris disability scale	√	√
Pain self-efficacy	√	√
Start back assessment tool	√	
MSK-HQ	√	√
Satisfaction		
Satisfaction survey		√

**Table 2 healthcare-09-00238-t002:** Logic model process.

Determents	Actions	Outputs	Outcome	Impact
**Intervention Characteristics** Relative advantage Adaptability Trialability Cost **Outer setting** Patients need and resources Cosmopolitanism Peer pressure External policy and incentives **Inner setting** Tension for change Goals and feedback Learning climate Culture **Characteristics of individuals** Knowledge and beliefs about intervention Self-efficacy **Process** Executing Engagement of formal implementation leaders	Limit referrals for MRIs for back pain to the spine clinic unless specific pathology is suspected. Train physiotherapists in: ○ Red-flag Screening ○ Screening using the STarT Back Screening Tool. Educate physiotherapists and spine surgeons about the biopsychosocial model, multi-disciplinary teamwork and effective communication. Patients triaged by physiotherapist for spine surgeon referral or physiotherapy only. “First point of contact physiotherapist” screeners refer to different specialists. Medium and high-risk patients receive both physical and psychologically informed physiotherapy.	Early identification of patients who need further services. All patients attending the spine clinic are screened for psychosocial risk factors using the STarT Back Screening Tool. All patients educated about back pain and self-management. Specialist referrals triaged according to risk. Improved team communication. Multidisciplinary cooperation. Team approach to rehabilitation adopting the biopsychosocial model.	25% fewer LBP referrals for MRIs. More efficient MRI referrals. More appropriate physiotherapy referrals. Reduced waiting times to access physiotherapy (≤ 30%). Reduced waiting time for spine surgeon consultation (≤ 30%) Patients’ increased confidence in self-management of LBP. Increased evidence-based practice. Increased confidence in clinical decision making (physiotherapists and spine surgeons).	Increased satisfaction among physiotherapy and spine clinic staff. Greater patient satisfaction with services. Reduction in the direct cost of care associated with LBP. More efficient use of tertiary care services. More people at work and fewer days lost to sickness/absences.

Abbreviations: LBP, lower back pain; MRI, magnetic resonance imaging.

## Data Availability

The datasets used and/or analysed during the current study are available from the corresponding author upon reasonable request.

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
