# Peer review of "Development of a Logic Model for a Programme to Reduce the Magnetic Resonance Imaging Rate for Non-Specific Lower Back Pain in a Tertiary Care Centre"

_healthcare, 2021, doi:10.3390/healthcare9020238_

Round 1
Reviewer 1 Report
This is protocol of important topic.
I suggest the investigators to provide the decision tree algorithms as figure to demonstrate better.
Please provide the information whether neurologist, neurosurgeon, and spine Orthopaedic will be involved in these processes? and in which stages?
Author Response
Thank you for allowing us to submit a revised draft of our manuscript titled " “Development of a logic model for a programme to reduce the magnetic resonance imaging rate for non-specific low back pain in a tertiary care centre”. We appreciate the time and effort that you have dedicated to providing your valuable feedback on our manuscript. We are grateful for your insightful comments on our paper. We have been able to incorporate changes to reflect most of the suggestions. We have highlighted the changes within the manuscript.

Reviewer 2 Report
Thank you for an interesting read.
In addition to Hartvigsen 2018, consider adding 1 (Nadine 2018) or both of the remaining articles from the Lancet LBP series. Also, elaborating (further) on the psychosocial consequences of current imaging practices (e.g. catastrophizing, dependency etc.) might improve the manuscript.
https://www.thelancet.com/series/low-back-pain
Aside from a few typos and the need for improving the visual layout of both tables, I have no further comments on the manuscript.
I look forward seeing the program put into action.
Author Response

(The authors gave the same response as above.)

Round 2
Reviewer 1 Report
all of my comments have been addressed